# Semi-Supervised Medical Image Segmentation via Knowledge Mining from Large Models

## Abstract

Large-scale vision models like SAM possess extensive visual knowledge, but their application to specialized tasks like medical image segmentation is often hindered by their general nature and the computational challenges associated with training and finetuning. Locally hosted small models such as U-Net++, designed for specific tasks, struggle with limited performance due to sparse labeled datasets. This study introduces a strategic knowledge mining method as a novel interaction mechanism between large and small models. Our method utilizes SAM's broad visual understanding to enhance the specialized capabilities of locally hosted small deep learning models. Specifically, we trained a U-Net++ model on a limited labeled dataset and extend its capabilities by converting outputs (masks) produced in unlabeled images into prompts, to extract relevant knowledge from SAM. This process not only harnesses SAM's generalized visual knowledge but also iteratively improves SAM's prediction to cater specialized medical segmentation tasks via U-Net++. The mined knowledge, serving as 'pseudo labels', enriches the training dataset, enabling the fine-tuning of the local network. Applied to the Kvasir SEG and COVID-QU-Ex datasets which consist of gastrointestinal polyp and lung X-ray images respectively, our proposed method consistently enhanced the segmentation performance on Dice by 3% and 1% respectively over the baseline U-Net++ model, when the same amount of labelled data were used during training (75% and 50% of labelled data). Remarkably, our proposed method surpassed the baseline U-Net++ model even when the latter was trained exclusively on labeled data (100% of labelled data). These results underscore the potential of knowledge mining to overcome data limitations in specialized models by leveraging the broad, albeit general, knowledge of large-scale models like SAM, all while maintaining operational efficiency essential for clinical applications. The code of our method is publicly available at this link.

## 1 Introduction

Segmentation is a crucial task in the medical domain with numerous downstream clinical applications, including disease diagnosis, treatment planning, and surgical outcome prediction (Guo et al., 2022; De Fauw et al., 2018). The evolution of deep learning has significantly enhanced medical segmentation capabilities, transitioning from lightweight models like U-Net to more complex and specialized architectures (Ronneberger et al., 2015; Hatamizadeh et al., 2021; Dumitru et al., 2023). Despite these advancements, the field still grapples with the challenge of limited access to large-scale, high-quality annotated datasets. These datasets commonly require trained professionals for manual labeling, which is labor-intensive, cost-inefficient, and are frequently unavailable due to privacy concerns.

The success of adapting large foundational models trained on large-scale datasets for some specific medical image analysis tasks not requiring pixel level annotations, such as disease classification, offers a promising approach to mitigate the problem of scarce data. These foundation models leverage broad and versatile training data on a variety of images, establishing robust feature representations that can be instrumental in various downstream medical image tasks. Previous efforts have focused on adapting models trained with techniques like DINO Oquab et al. (2024), MAE He et al. (2021), or other self-supervised pre-trained methods, which are mainly suitable for classification tasks.

The recent introduction of the Segment Anything Model (SAM) Kirillov et al. (2023) offers a promising solution for the specific adaptation to medical image segmentation. SAM's potential stems from its training on a dataset comprising one billion natural image-mask pairs, which equips it with a robust foundation for diverse segmentation tasks.

Given SAM's strong performance and great generalizability on natural images, recent research has evaluated SAM's zero-shot performance on medical datasets, including tests on CT, MRI, pathological, and various other modalities (Ji et al., 2022; Zhang & Wang, 2023; Deng et al., 2023). Despite its potential, a performance gap persists between SAM and state-of-the-art segmentation methods, attributable to its training solely on natural image datasets. This suggests that SAM would benefit from further adaptation or guidance.

Addressing the challenge of zero-shot performance of SAM in medical domains, researchers have attempted to fine-tune SAM on medical datasets. For example, MedSAM fine-tunes SAM on comprehensive and diverse medical image datasets (Ma et al., 2024). Other approaches have adopted parameter-efficient fine-tuning for improved training efficiency (Zhang & Liu, 2023; Wu et al., 2023). However, these methods still require high-quality prompts during inference due to SAM's underlying structure. To avoid the need for prompting during inference, some methods have chosen to guide SAM through automatic prompting using guiding points and bounding boxes by incorporating the YOLO structure or framing it as a localization task (Pandey et al., 2023; Lei et al., 2023). However, these adaptation-based methods invariably lead to large models, hindering their operational efficiency and practicality in clinical settings.

To achieve the operational efficiency required while attaining good performance under sparsely labeled medical datasets, we propose a strategic knowledge mining method as a novel interaction mechanism between large and small models. By training a lightweight U-Net++ Zhou et al. (2018) model on a limited-labeled dataset, we then use it to guide the generalist SAM in generating pseudo labels for unlabeled data, which can be further used to boost the lightweight model's performance. This novel interaction not only facilitates learning on scarcely labeled datasets, but also mines and inject the domain-specific knowledge of SAM into the U-Net++ model. During inference, our method also offers a balance between operational efficiency and accuracy. When operational efficiency is a top priority, lightweight U-Net++ model can allow fast inference. On the other hand when higher accuracy is needed, SAM can be involved to take in summarized prompt from U-Net++ prediction, such that the result can be refined with improved accuracy. Additionally, since we do not propagate gradients back to SAM, the training process is also memory-efficient compared to directly fine-tuning SAM on medical datasets.

We validated our proposed technique using the the Kvasir-Seg Jha et al. (2020) dataset, demonstrating superior performance when our method was trained on partially labeled data, compared to training directly on the full dataset in a supervised setting. Furthermore, we showed that the proposed method complements existing methods, and by incorporating self-supervised learning (SimCLR) or MedSAM, the lightweight U-Net++ can further improve their performance.

To summarize, our contributions are as follows:

1. We propose a strategic knowledge mining method as a novel interaction mechanism between large and small models, which facilitates data-efficient segmentation.

2. We tested different types of visual prompts generated by the lightweight student model and identified the most effective prompting techniques (point and bounding box).

3. We demonstrated that the proposed method could further benefit from domain-specific fine-tuned SAM models and other self-supervised techniques.

## RELATED WORKS

The scarcity of large-scale medical datasets has led to the development of various approaches to address this issue. Our method is broadly related to the research direction of semi-supervised learning, knowledge mining through large-small model interaction, and adapting SAM for medical image segmentation.

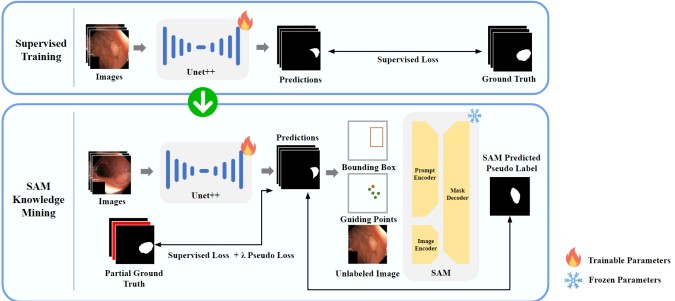

Figure 1: Knowledge mining procedure for SAM. The U-Net++ is first trained in a supervised setting and directly adopted for SAM knowledge mining. The red square implies unlabeled images. The fire and snowflake icons indicate trainable and frozen modules, respectively. The respective supervised loss and pseudo loss are illustrated in Section 2.5.

### Semi-supervised Semantic Segmentation

Semi-supervised learning is a popular approach for addressing the scarcity of labeled data. It leverages both labeled and unlabeled data to augment the dataset, thereby improving model performance. One of the most related approaches to our method is pseudo-labeling, which generates pseudo masks for unlabeled data to increase the number of training samples. This approach focuses on generating reliable pseudo labels.

Pseudo-labeling was initially proposed for classification tasks, where the argmax of the softmax prediction is treated as a pseudo label (Lee, 2013). This methodology was later adapted for semantic segmentation by applying a threshold to model's predictions, converting them into binary threshold predictions, used as pseudo labels (Feng et al., 2022). To enhance the quality of pseudo labels, Yao et al. (2022) has incorporated confidence ratings, where a confidence score is assigned to each pixel of the predicted mask by calculating the pixel-wise variance between predictions on the original and transformed images. Li et al. (2021) uses an exponential moving average on pseudo labels, continuously updating them by combining previous and current pseudo labels to reduce noise and inconsistency. PseudoSeg performed both strong and weak augmentation on the same input images, using the weakly augmented image as the pseudo label (Zou et al., 2021). Unlike these methods, which rely on the current model predictions and use additional techniques to clean the predicted masks, our method consults SAM as the generator of pseudo labels. This strategy avoids the issue of unreliable learning from incorrect answers that can degrade training results.

### Knowledge Mining through Model Interaction

In the broader context of knowledge mining through model interaction, our research connects with the field of LLM-aided visual reasoning. This field involves multi-modal models that interact with specialized models, such as captioning or detection models, to refine their outputs. However, these interactions are typically conducted in a zero-shot manner, lacking feedback loops where the model is trained on the extracted knowledge and without producing a lightweight model for efficient inference (Yang et al., 2023b; Wang et al., 2023; Yang et al., 2023a). Specifically, our method focuses on knowledge mining from SAM, in the medical domain. SAMAug-C is an example where SAM-predicted masks are combined with original images for classification (Gu et al., 2024). The work most relevant to ours is Li et al. (2024), which employs SAM as a pseudo-label generator for semi-supervised learning guided by a pre-trained SS-Net model. Our method differs as we allow iterative evolution of visual prompts generated from unlabeled data. Additionally, our approach not only results in a lightweight U-Net++ model but also allows SAM to be capable of producing fine-grained predictions in medical images by leveraging information from a U-Net structure. Another similar approach to ours is SAMAug (Zhang et al., 2023b). Both our method and SAMAug utilize an external model to prompt SAM. However, SAMAug utilizes a frozen Visual Saliency Transformer to generate a saliency map from which point prompts are randomly sampled, whilst our method employs a specifically trained U-Net++ model on the target dataset, selecting point prompts based on the highest probability within the predicted mask. Thus, our method allows more efficient and

dataset-specific inference. Furthermore, our work extends beyond previous efforts by prioritizing two critical aspects: (1) achieving high performance even with limited labeled data, and (2) ensuring operational efficiency.

### ADAPTING SAM FOR MEDICAL IMAGE SEGMENTATION

As part of our main contributions, our method adapts SAM for medical image segmentation. After training, our method allows the U-Net++ to act as an automatic prompter, enabling SAM to infer on medical images, thus adapting SAM for medical segmentation. Given the challenges in SAM's zero-shot performance on medical image segmentation, some researchers have chosen to directly fine-tune SAM on versatile medical datasets. MedSAM fine-tunes SAM on large scale medical image datasets, using a variety of modalities (Ma et al., 2024). For more memory-efficient fine-tuning, SAMed adopts parameter-efficient fine-tuning (PEFT) techniques like Low-Rank Adaptation, while SAM-SA adapts SAM for 3D medical image segmentation using prompt-conditioned adaptation (Wu et al., 2023; Zhang & Liu, 2023). Although effective in adapting SAM for medical datasets, these methods still require high-quality prompts during inference, which is commonly impractical due to reliance on involving trained medical professionals in the loop.

To eliminate the need of prompting during inference, a promising direction is to learn a prompt embedding that can be directly utilized by the prompt encoder. All-in-SAM trains a custom prompt embedding using SAM-derived image embeddings and high-frequency data (Cui et al., 2023). AutoSAM employs a harmonic Dense-net that takes the image as input and outputs a mask prompt for the mask decoder (Shaharabany et al., 2023). DeSAM replaces the mask decoder with a Prompt Relevant IOU Module (PRIM) and a Prompt Invariant Mask Module (PIMM). PIMM processes the image and prompt embeddings from random points together with a mask embedding through cross-attention, which is then concatenated with the image embedding and decoded by PIMM (Gao et al., 2023). SAM-Path introduces a pathological encoder parallel to SAM's image encoder and learns a class embedding prompt for each pathological class (Zhang et al., 2023a). Although these methods are effective in adapting SAM for medical image segmentation, they still face operational efficiency challenges during inference, as SAM itself is large.

## 2 METHODOLOGY

Consider a partially labeled source dataset, $\mathbb{S}$, with the labeled subset $\{\mathbb{X}_L, \mathbb{Y}_L\}$ and abundant unlabeled data, $\{\mathbb{X}_U\}$ from the same modality, we aim to design a strategy that mines medical domain specific knowledge from SAM to generate pseudo label for medical image segmentation on the unlabeled data to boost the dataset and improve the overall performance by enhancing the interaction between large-small models.

### 2.1 REVISITING SEGMENT ANYTHING

The Segment Anything Model is a foundational model for natural image segmentation, consisting of three components: an image encoder, a prompt encoder, and a lightweight mask decoder. The image encoder employs a MAE pre-trained vision transformer, which encode any input image into an image embedding. The prompt encoder accepts three types of input: guiding points, bounding boxes, and mask. Guiding points and bounding boxes are encoded into a sparse embedding, which is the summation of a trained embedding and the prompt location's positional encoding. Mask prompt are encoded into a dense embedding. To generate the segmentation mask, the mask decoder first adds the dense and image embeddings point-wise, then enhances the features by interacting with the sparse embedding through two cross-attention layers to decode the final segmentation masks.

### 2.2 SEMI-SUPERVISED KNOWLEDGE MINING

In medical imaging, where specific modalities commonly involve largely sparse labels, SAM's robust generalization capability is invaluable for mining domain-specific knowledge. Given a limited labeled dataset, knowledge mining is best performed in a semi-supervised manner. To extract the desired domain-specific knowledge from a large segmentation model like SAM, we first train a lightweight U-Net++ model on a sparsely labeled dataset, which acts as a domain-specific "student"

model. We leverage the "student" model to generate predicted masks on unlabeled data. These masks are then transformed into guiding points and bounding box information, which are subsequently used to prompt the generalist SAM model which acts as the "teacher" model. The generalist "teacher" model leverage its extensive natural image knowledge, producing more accurate results over-time (during training), which then serve as pseudo labels for subsequent training. The U-Net++ model is subsequently trained on these pseudo labels, data it has not encountered before, leading to improved performance. Although the lightweight U-Net++ model does not perform perfectly on most cases due to sparse labeling during training, it can still effectively guide SAM. This is thanks to SAM's rich natural image knowledge, which allows for effective domain-specific knowledge mining with minimal domain-specific prompting. Additionally, the SAM model remains frozen throughout the entire process, ensuring that we are only mining knowledge from it. This prevents any inaccuracies in prompts that could potentially poison SAM's knowledge base.

## 2.3 GENERATION OF PSEUDO LABELS

To generate reliable pseudo labels, we first train a lightweight U-Net++ model on the labeled subset $\{\mathbb{X}_L, \mathbb{Y}_L\}$ until convergence. Optionally, the lightweight U-Net++ model can be pre-trained with self-supervised learning methods on both the labeled and unlabeled subsets $\{\mathbb{X}_L, \mathbb{X}_U\}$ to boost the performance. Once the lightweight U-Net++ model has been sufficiently trained, we can proceed to create pseudo labels. Using the trained lightweight model, we make initial predictions on the unlabeled subset $\{\mathbb{X}_U\}$. These initial predictions may be inaccurate and not yet suitable to be used as pseudo labels for future training. Therefore, we consult SAM using the information from the lightweight model's predictions to further improve segmentation. SAM can accept three types of prompts: guiding points, bounding boxes, and mask, enabling us to incorporate medical domain-specific information extracted from the lightweight model's predictions into SAM through these prompts. Specifically, the prompts are been extracted as follows:

**Guiding Points Prompt**    To derive guiding points prompt, we need to sample $x$ points from the image that best describes the location of desired object. We use the predictions from the lightweight model to identify these $x$ points. Each pixel's value in the model's predictions indicates the estimated probability of that pixel belonging to the target object. Therefore, we propose sampling $x$ points from the pixels with the highest probabilities. This approach allows us to efficiently represent the information in the predicted mask using a limited number of points. In cases where more than $x$ points share the maximum probability, we randomly select 'x' points from this set to generate the point prompt.

**Bounding Box Prompt**    Given that the point prompts are selected based on the highest probabilities and that a model is generally more confident near the center of an object, the point prompt typically describes the center of the predicted mask. However, point prompts alone fail to convey the desired extent of the segmentation, leaving SAM unaware of the boundaries beyond the center point. This poses a challenge due to SAM's inherent design, which generates three distinct masks representing the whole, part, and subpart of an object. Hence, it is crucial to provide information on the size of the desired object for SAM to produce high-quality masks. Following experimentation, we choose to use the outer box of the predicted mask, thresholded at $0.5$, as the bounding box prompt, as it effectively captures the prediction mask entirely.

**Mask Prompt**    We chose to neglect the mask prompt when prompting SAM, as it imposes overly strict constraints, specifically in terms of the point-wise addition of embeddings. Any inaccuracies in the lightweight model's mask predictions could be amplified if passed directly to SAM. While incorporating mask prompts may improve the separation of small masks, this approach introduces a trade-off, as the stricter constraints could negatively affect overall performance. We observed this trade-off during preliminary experiments and have included an ablation study to explore its impact further.

In summary, our method uses points and bounding box prompts as input to SAM. By utilizing these prompting strategies, we can effectively inject domain- specific information from the lightweight model into SAM, thereby producing reliable pseudo labels that ultimately enhanced the dataset. This step was critical to improve the overall method performance.

### 2.4 PSEUDO LABEL GENERATION SCHEDULING STRATEGIES

For the SAM knowledge mining process, we introduce two pseudo label generation scheduling strategies: one-time generation and continuous generation.

**One-Time Pseudo Label Generation**   For one-time pseudo label generation, pseudo labels are generated only once after the supervised training. When the supervised training of the lightweight model is completed, we perform inference on the unlabeled data $\{\mathbb{X}_U\}$ and convert the inferred predictions to SAM prompts. Base on these prompts, SAM produces batches of predictions, that are treated as pseudo labels $\{\mathbb{Y}_U^{psedo}\}$. The lightweight model is subsequently trained with the completed dataset, $\{\mathbb{X}_L \cup \mathbb{X}_U, \mathbb{Y}_L \cup \mathbb{Y}_U^{psedo}\}$. This approach allows fast training, as SAM is consulted only once for the entire set of unlabeled data during training.

**Continuous Pseudo Label Generation**   In continuous pseudo label generation, a pseudo label is generated each time an unlabeled data point is revisited. When the lightweight model is trained and ready to be applied to the unlabeled data, we generate a pseudo label on-the-fly. For each unlabeled data point, we first infer using the lightweight model, then predict with SAM using the extracted prompts. The resulting pseudo label is directly compared with the lightweight model's prediction for loss evaluation. Although SAM inference necessitates additional training time, it enables the pseudo labels to evolve in tandem with the lightweight model, enhancing their quality.

### 2.5 LOSS FUNCTION

For both supervised learning and SAM knowledge mining phases, we employed a widely adopted loss function, consisting of a weighted combination of binary cross entropy loss and dice loss (Ma et al., 2024; Ahmed et al., 2020). BCE loss helps with curve smoothing, while Dice loss addresses class imbalance, leveraging the strengths of both. Let $B$ and $B'$ be the number of labeled and unlabeled data in a batch, respectively. The parameter $k$ is a hyper parameter that controls the weighting between the BCE and Dice losses. Based on empirical results from our experiments, we set $k = 0.2$ for optimal performance. Therefore, we have the supervised loss defined as:

$$L = \frac{1}{B} \sum_B (L_{Dice} + kL_{BCE})$$

During SAM knowledge mining phase, we down-weighted the pseudo label's loss by a scalar factor $\lambda$, acknowledging the inherent uncertainty compared to ground truth segmentation masks. In our experiments, we set $\lambda$ to be 0.25. The total loss then becomes:

$$L = \frac{1}{B} \sum_B (L_{Dice} + kL_{BCE}) + \lambda \frac{1}{B'} \sum_{B'} (L_{Dice}^{pseodo} + kL_{BCE}^{pseodo}),$$

where $L_{Dice}^{pseodo}$ and $L_{BCE}^{pseodo}$ represent the Dice and BCE losses, respectively, calculated with pseudo labels replacing the ground truth.

### 2.6 DATASET

We have used Kvasir-SEG Jha et al. (2020) and COVID-QU-Ex Tahir et al. (2021); Chowdhury et al. (2020) datasets for training and evaluation.

**Kvasir-SEG Dataset**   The Kvasir-SEG Dataset is a large-scale dataset of gastrointestinal polyp images and its corresponding segmentation masks. Kvasir-SEG contains 1,000 segmented polyp images with varied resolutions ranging from $332 \times 487$ to $1920 \times 1072$. We randomly split the dataset into 80%, 10%, and 10% subsets for training, validation, and testing. Although there are similar large-scale datasets of gastrointestinal images without labeled segmentation masks, such as Kvasir Pogorelov et al. (2017) and HyperKvasir Borgli et al. (2020), the unlabeled data in these datasets either have different target tasks then polyp segmentation or already existed in the Kvasir-SEG, preventing their use to augment our unlabeled dataset.

Table 1: Performance on the Kvasir-SEG dataset. Under different percentage of training data, we compared supervised training on labeled data, our semi-supervised training on all data, and the integration of SimCLR and MedSAM. We also included Unet++ trained on $100\%$ labeled data as a baseline. The gold- and blue-highlighted item indicates the overall best performance within the relevant split ($75\%$ and $50\%$, respectively).

| METHODS | IOU (AVG±STD) | DICE (AVG±STD) |
|---|---|---|
| **Labeled/Unlabeled Split (100% Labeled)** | | |
| Supervised Training on Labeled Data | $0.649 \pm 0.015$ | $0.753 \pm 0.015$ |
| **Labeled/Unlabeled Split (75% Labeled)** | | |
| Supervised Training on Labeled Data | $0.617 \pm 0.012$ | $0.722 \pm 0.010$ |
| Continuous Pseudo Label Generation | $0.658 \pm 0.005$ | $0.756 \pm 0.003$ |
| One-Time Pseudo Label Generation | $0.642 \pm 0.016$ | $0.743 \pm 0.016$ |
| SimCLR + Supervised Training on Labeled Data | $0.637 \pm 0.002$ | $0.739 \pm 0.003$ |
| SimCLR + Continuous Pseudo Label Generation | $0.647 \pm 0.016$ | $0.747 \pm 0.014$ |
| SimCLR + One-Time Pseudo Label Generation | $0.652 \pm 0.015$ | $0.754 \pm 0.013$ |
| MedSAM + Continuous Pseudo Label Generation | $0.655 \pm 0.013$ | $0.756 \pm 0.016$ |
| MedSAM + One-Time Pseudo Label Generation | $0.649 \pm 0.039$ | $0.749 \pm 0.035$ |
| **Labeled/Unlabeled Split (50% Labeled)** | | |
| Supervised Training on Labeled Data | $0.575 \pm 0.021$ | $0.680 \pm 0.023$ |
| Continuous Pseudo Label Generation | $0.607 \pm 0.032$ | $0.708 \pm 0.030$ |
| One-Time Pseudo Label Generation | $0.607 \pm 0.020$ | $0.706 \pm 0.017$ |
| SimCLR + Supervised Training on Labeled Data | $0.561 \pm 0.053$ | $0.670 \pm 0.044$ |
| SimCLR + Continuous Pseudo Label Generation | $0.595 \pm 0.054$ | $0.696 \pm 0.050$ |
| SimCLR + One-Time Pseudo Label Generation | $0.572 \pm 0.066$ | $0.678 \pm 0.059$ |
| MedSAM + Continuous Pseudo Label Generation | $0.604 \pm 0.020$ | $0.706 \pm 0.015$ |
| MedSAM + One-Time Pseudo Label Generation | $0.605 \pm 0.029$ | $0.704 \pm 0.028$ |

**COVID-QU-Ex Dataset**  The COVID-QU-Ex dataset is a dataset designed for lung segmentation from X-ray images of COVID-19 infected, non-COVID infected, and normal lungs. Specifically, we used the "COVID-19 Infection Segmentation Data" from the COVID-QU-Ex dataset, which includes 3,962 image-mask pairs for lung segmentation. The dataset is pre-split into 1864/1166/932 pairs for training, validation, and testing, respectively.

## 2.7 EVALUATION METRICS

To measure the performance of the predicted masks, we followed the suggested metrics stated in the Kvasir SEG and used Intersection Over Union (IOU) and Dice similarity coefficient (DICE) to quantitatively evaluate the segmentation results. Both metrics are region-based, designed to measure the overlap between ground truth masks and predicted segmentation results, and are defined as:

$$\text{Dice}(y, \hat{y}) = \frac{2|y \cap \hat{y}|}{|y| + |\hat{y}|}, \quad \text{IoU}(y, \hat{y}) = \frac{|y \cap \hat{y}|}{|y \cup \hat{y}|}$$

where $y$ represents the ground truth mask, $\hat{y}$ represents the predicted segmentation mask. These metrics provide a comprehensive evaluation of the segmentation performance by considering both the degree of overlap and boundary alignment between the predicted and ground truth masks.

## 2.8 TRAINING PROTOCOL AND EXPERIMENTAL SETTING

To test the effectiveness of our method on different numbers of unlabeled and labeled data, we divided the training subset of each dataset into a labeled set and an unlabeled set, where the labels in

Table 2: Performance results on the COVID-QU-Ex dataset. Results are presented for each Labeled/Unlabeled split, comparing the supervised baseline and our semi-supervised method under both continuous and one-time pseudo label scheduling. The baseline trained on 100% labeled data is also included. The gold- and blue-highlighted item indicates the overall best performance within the relevant split (75% and 50%, respectively).

| METHODS | IOU (AVG±STD) | DICE (AVG±STD) |
|---|---|---|
| **Labeled/Unlabeled Split (100% Labeled)** | | |
| Supervised Training on Labeled Data | $0.897 \pm 0.010$ | $0.944 \pm 0.006$ |
| **Labeled/Unlabeled Split (75% Labeled)** | | |
| Supervised Training on Labeled Data | $0.883 \pm 0.002$ | $0.936 \pm 0.002$ |
| Continuous Pseudo Label Generation | $\mathbf{0.900 \pm 0.003}$ | $\mathbf{0.945 \pm 0.002}$ |
| One-Time Pseudo Label Generation | $0.895 \pm 0.003$ | $0.943 \pm 0.002$ |
| **Labeled/Unlabeled Split (50% Labeled)** | | |
| Supervised Training on Labeled Data | $0.880 \pm 0.007$ | $0.933 \pm 0.004$ |
| Continuous Pseudo Label Generation | $\mathbf{0.898 \pm 0.007}$ | $\mathbf{0.944 \pm 0.004}$ |
| One-Time Pseudo Label Generation | $0.896 \pm 0.004$ | $0.943 \pm 0.003$ |

the unlabeled set were dropped to mimic unlabeled data. The ratios of labeled to unlabeled sets were designed to be 100%/0%, 75%/25%, 50%/50%, respectively. The 100%/0% distribution represents supervised learning on the original dataset, establishing the upper bound of the selected lightweight model. The remaining splitting ratios are designed to test our method's performance at scenarios with different levels of labeled and unlabeled data.

Moreover, since our approach aims to extract SAM's knowledge into a lightweight model for operational efficiency, we aim to compare our method with other pre-training methods, using the same lightweight model. Comparison with other large state-of-the-art models is precluded because lightweight models such as U-Net++ have inherent limitations and/ or performance upper bounds, making it unfair to compare them against complex and large models. For a comprehensive evaluation, we incorporated SimCLR in our method, which is a popular self-supervised learning method (Chen et al., 2020). We also incorporated MedSAM to determine if a domain-specific adapted SAM would further improve the lightweight model's performance in our combined method (Ma et al., 2024).

For ablation studies, we tested the setting of training only with point prompts or bounding box prompts on the "75% Labeled" split to examine the significance of both prompts. We alsopresent the result of our method when incorporating the mask prompt, demonstrating the trade off stated in Mask Prompt paragraph of Section 2.3.

## 3 QUANTITATIVE RESULTS

### 3.1 RESULTS ON KVASIR-SEG DATASET

Table 1 presents the performance results of the trained U-Net++ under different data splits and pseudo label scheduling strategies for Kvasir-SEG dataset. The "Supervised Training on Labeled Data" row with a Labeled/Unlabeled Split of "100% train" reflects the U-Net++'s performance through supervised training on the original dataset, serving as an upper bound performance of U-Net++.

When examining the result of our methods across different different Labeled/Unlabeled split, we can notice that both IOU and DICE scores consistently show that methods utilizing pseudo labels (with either pseudo label generation strategies) outperform the "Supervised Training on Labeled Data" approach within each split. This suggests that leveraging SAM for pseudo label generation enhances the model's segmentation accuracy beyond what is achievable with solely supervised training on labeled data. Additionally, we have observed that the continuous updates works better the majority

Table 3: Ablation study on different prompting approaches. The gold-highlighted item indicates the overall best performance

| METHODS | IOU (AVG±STD) | DICE (AVG±STD) |
|---|---|---|
| **Labeled/Unlabeled Split (75% Labeled)** | | |
| Supervised Training on Labeled Data | $0.617 \pm 0.012$ | $0.722 \pm 0.010$ |
| Continuous Pseudo Label Generation | $\mathbf{0.658 \pm 0.005}$ | $\mathbf{0.756 \pm 0.003}$ |
| One Time Pseudo Label | $0.642 \pm 0.016$ | $0.743 \pm 0.016$ |
| Continuous Pseudo Label Generation (Box) | $0.632 \pm 0.019$ | $0.732 \pm 0.017$ |
| One-Time Pseudo Label Generation (Box) | $0.638 \pm 0.014$ | $0.739 \pm 0.014$ |
| Continuous Pseudo Label Generation (Points) | $0.587 \pm 0.032$ | $0.695 \pm 0.029$ |
| One-Time Pseudo Label Generation (Points) | $0.612 \pm 0.035$ | $0.713 \pm 0.033$ |
| Continuous Pseudo Label Generation (Points, Box, Mask) | $0.637 \pm 0.013$ | $0.738 \pm 0.015$ |
| One-Time Pseudo Label Generation (Points, Box, Mask) | $0.625 \pm 0.029$ | $0.729 \pm 0.029$ |

of the time. This aligns with our initial design rationale, where the ability of continuous pseudo label generation dynamically updates and improves labels as the U-Net++ model evolves, leading to superior performance compared to one-time pseudo label generation.

Furthermore, a substantial improvement is observed in the 75% Labeled split, where both continuous pseudo label and one-time pseudo label generation scheduling surpass the "Supervised Training on Labeled Data" approach by 3%. Continuous pseudo label generation, in particular, even surpasses the baseline of purely supervised training on the original dataset. This improvement can be attributed to the continuous refinement of pseudo labels, which effectively augments the training data and enhances the model learning process.

When combined with other methods, our approach maintains strong performance. Across all splits, SimCLR combined with our method consistently achieves higher scores than training solely on labeled data or pre-training with SimCLR and finetuning with labeled data. Training our methods with MedSAM still shows noticeable improvement over purely supervised training. Compared to training with SAM, MedSAM performs better with One-Time Pseudo Label (e.g., see the "75% split"). The improved performance is likely due to MedSAM's prior fine-tuning on medical data, resulting in more accurate initial pseudo labels compared to SAM.

## 3.2 RESULTS ON COVID-QU-EX DATASET

Table 2 presents the results on the COVID-QU-Ex dataset. The performance trend mirrors earlier results: improved performance compared to supervised training on each split, with continuous pseudo labeling outperforming the one-time approach. Notably, the performance of our method on the 50% split trained with continuous SAM pseudo label clearly outperforms the fully supervised learning baseline method.

These results underscore the effectiveness of our approach and its capability to integrate complementary methods, enhancing the segmentation accuracy of lightweight models like U-Net++ on sparsely labeled datasets.

## 3.3 ABLATION STUDY

Table 3 presents the results of the ablation study conducted on the Kvasir-SEG dataset, focusing on the usage of different prompt types in the SAM framework. The study was conducted using the "75% Labeled" split. Training with only points prompt resulted in a significant drop in the segmentation performance compared to training with bounding box prompt. Specifically, when the bounding box prompt was omitted with continuous pseudo label generation, the segmentation accuracy degraded to a level worse than that achieved by supervised training alone. This suggests that without the bounding box prompt, SAM struggles to determine whether the user wants the whole, part, or subpart of an object, leading to degraded pseudo label quality that further impacts

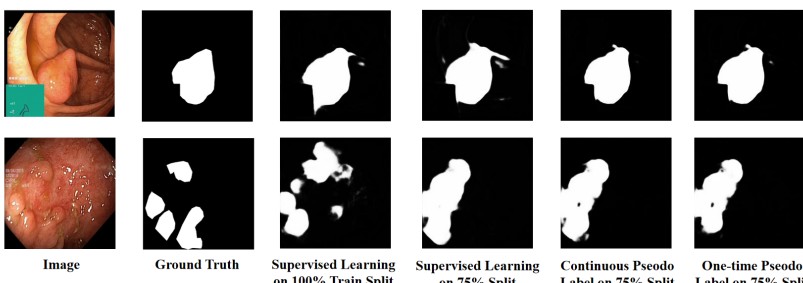

Figure 2: Sample results on the Kvasir-SEG testing set for qualitative analysis.

model performance. Similarly, when operating with only the bounding box prompt, the performance was inferior to using both prompts. Hence, ablation results confirm the necessity of both point and bounding box prompts for optimal pseudo label generation using SAM, as removing either adversely affects performance.

To assess the necessity of the mask prompt, we have presented the results of our method using all three types: point, bounding box, and mask prompts, as shown in the last two rows of Table 3. Our findings indicate that although the model performance remains strong and optimal (versus the "Supervised Training on Labeled Data" results), including the mask prompt resulted in a decline in performance compared to using both point and bounding box prompts. This decline was consistent in both the continuous and one-time pseudo label scenarios, suggesting that the inclusion of the mask prompt may introduce challenges in some instances that outweigh its potential benefits. The inclusion of masks may add constraints in the robustness and generalizability of the SAM model, particularly in instances where the masks are less accurate or reflect particularities in the image semantics and brightness.

### 3.4 QUALITATIVE ANALYSIS

In Figure 2, we present samples of predicted results on the Kvasir-SEG test set using the lightweight model trained on "100% Labeled" and "75% Labele" splits with different pseudo label generation schedules. Upon observing the results of fully supervised learning on the entire training set, we note that the predicted masks often appear blurred and extend beyond the actual polyp regions, incorporating extra parts. In contrast, both continuous and one-time pseudo labeled methods produce more compact masks without additional sections. This improvement can be attributed to SAM's general knowledge that optimally guides pseudo label generation. However, in the "75% Labeled" split scenario (second row), we observe instances where the fully supervised lightweight model struggles to accurately delineate multiple separate polyp masks. This challenge affects the performance of our method. While a unified mask is still favored, SAM enables both scheduling modes to attempt segmentation of distinct small polyp masks, visible as small black areas in between structures. This suggests potential for further advancements beyond the current state.

## 4 CONCLUSION

This study demonstrates the successful adaptation of SAM's generalized visual knowledge for specialized medical image segmentation. By utilizing mined knowledge as 'pseudo labels,' we fine-tuned a local network, achieving a $> 3\%$ performance improvement on Kvasir-SEG compared to both baseline and fully supervised U-Net++. Our proposed method also outperformed other pre-trained models (SimCLR, MedSAM) when these were combined with our U-Net++. Consistent results were observed in the COVID-QU-Ex dataset, with continuous pseudo-labeling outperforming the one-time approach. An ablation study confirmed the necessity of both point and bounding box prompts. These findings highlight the potential of knowledge extraction to overcome data limitations in specialized models by leveraging the vast knowledge of large-scale models like SAM, while maintaining operational efficiency essential for clinical applications.

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

## A APPENDIX

### A.1 TRAINING PROTOCOL

**Data Augmentation** For data augmentation, we employed vertical and horizontal flips, rotation, and transpose with a probability of 0.5. To accommodate varying image sizes, we first re-scaled the images such that the shortest side was 224 pixels and used center cropping to ensure all images were sized at $3 \times 224 \times 224$.

**Training Details** For the lightweight model, we used U-Net++ model with a resnet34 as encoder (Zhou et al., 2018). During the supervised learning phase, the U-Net++ model was optimized using the Adam optimizer ($\beta_1 = 0.9$, $\beta_2 = 0.999$) with an initial learning rate of $5 \times 10^{-5}$ (Loshchilov & Hutter, 2019). The model was evaluated on the validation set at each epoch, and the learning rate was reduced by a factor of 0.5 if the validation loss did not decrease for 3 epochs. The minimum learning rate was set to $1 \times 10^{-7}$. Training was early stopped if the validation loss did not decrease for 10 consecutive epochs. We used a batch size of 8 for training on a T4 GPU. A detailed overview of the hyperparameters is provided in Table 4.

Table 4: Training Setting

| CONFIG | VALUE |
|---|---|
| optimizer | Adam |
| base learning rate | 5e-5 |
| weight decay | 0 |
| optimizer momentum | $\beta_1, \beta_2 = 0.9, 0.999$ |
| batch size | 8 |
| learning rate schedule | ReduceLROnPlateau |
| learning rate schedule mode | min |
| learning rate schedule patience | 3 |
| learning rate schedule factor | 0.5 |
| learning rate schedule min learning rate | 1e-7 |
| early stop epochs | 10 |
| training epochs | 100 |

