# OpenReview forum: "Semi-Supervised Medical Image Segmentation via Knowledge Mining from Large Models"
_ICLR.cc/2025/Conference — ICLR 2025 Conference Withdrawn Submission_

### Official Review · Reviewer_pfrE · 2024-10-31

**Soundness:** 3
**Presentation:** 3
**Contribution:** 3
**Rating:** 3
**Confidence:** 5

**Summary:**

This paper proposed a knowledge distillation model from a large frozen SAM model and a local trainable U-Net ++ model for semi-supervised medical image segmentation. The key idea is to leverage the SAM model to refine the pseudo labels and use both true and pseudo labels to train the model. Overall, the designs are reasonable and the segmentation performance is improved by using such designs on two public datasets.

**Strengths:**

The performance is improved a lot and even better than the fully-supervised models with all labeled data.

SAM is a powerful model and using this to refine the results is reasonable and should be able to improve the performance of semi-supervised segmentation.

**Weaknesses:**

Figure 1:
Figure 1 is too small to effectively show the core contributions of the paper. For instance, why is the supervised training placed at the beginning? If its role is to generate pseudo labels, it could be minimized to better highlight other key contributions such as how to generate the prompts from the pseudo labels.

Backbone Selection:
The choice of U-Net++ as the backbone requires further justification, as nnUNet is widely regarded as the backbone for most medical image segmentation tasks. Additionally, SAM has demonstrated superior performance compared to MedSAM from the experiments. Exploring other medical SAM backbones, such as SegVol, could potentially improve performance and emphasize the relevance of medical segmentation tasks.

[1] SegVol: Universal and Interactive Volumetric Medical Image Segmentation

Semi-Supervised Settings:
In the experiments, 75% and 50% labeled data were used for the model training. However, this setup is not optimal, as the baseline model already achieves strong performance with this level of labeled data. Consequently, the gains from semi-supervised learning appear marginal, which may limit the significance of the proposed approach. Also, please include several typical semi-supervised works for clear comparisons.

Medical Tasks and Clinical Motivation:
The paper focuses on segmenting polyps and lungs, but extending the analysis to other modalities, such as MRI or CT, and targets like small lesions or low-contrast regions, would further highlight its clinical relevance. This paper shows potential for using SAM to refine pseudo-labels, but a more general verification is necessary to support its general application in medical image segmentation.

Continuous pseudo label refinement could achieve the best performance but it costs more. An interesting study is to find a more general upper bound of the performance and please explain why the model achieves better performance than the upper bound.

**Questions:**

See the above weaknesses.
Overall, please
[1] Improve Fig.1
[2] Add more semi-supervised settings, select more backbones, and include more tasks
[3] Highlight the clinical motivation.

---

### Official Review · Reviewer_bPWd · 2024-11-03

**Soundness:** 1
**Presentation:** 2
**Contribution:** 1
**Rating:** 3
**Confidence:** 4

**Summary:**

This paper introduces a semi-supervised method leveraging the Segment Anything Model (SAM) to generate pseudo labels for training a lightweight U-Net++ segmentation model on limited labeled medical datasets. By using efficient point and bounding box prompts, the method extracts general visual knowledge from SAM, improving U-Net++'s performance without requiring SAM fine-tuning. The method also compares continuous and one-time pseudo labeling, finding continuous updates more beneficial. An ablation study underscores the effectiveness of the chosen prompting strategies. This efficient method enhances segmentation accuracy, making it well-suited for clinical applications with minimal labeled data.

**Strengths:**

+ The code is provided in the supplementary materials, which enhances the reproducibility of this work.

+ The approach of using SAM, prompted by segmentation models' pseudo labels to refine those labels and subsequently improve segmentation models, is logical and likely effective.

+ The authors investigated several strategies for generating pseudo labels (e.g., continuous vs. one-time) and enhancing segmentation models (e.g., using SimCLR, MedSAM). Notably, models trained on 75% labeled data achieved comparable or even superior performance to models trained on 100% labeled data.

+ Various methods for prompting SAM, including points, bounding boxes, and masks, were also explored.

+ The report is clearly presented, with all experimental settings detailed in the tables.

**Weaknesses:**

- **Lack of pseudo-label comparison:** The study does not quantify how pseudo labels generated by U-Net++ compare to those generated by SAM. This comparison against ground truth would clarify their relative effectiveness.

- **Significantly lower performance compared to existing benchmarks:** [The top reported DSC on Kvasir-SEG](https://arxiv.org/pdf/2311.02239v1) is 0.95, as seen in recent literature here, while the paper reports a DSC of 0.649. This substantial discrepancy raises concerns about the model's performance, choice of hyperparameters, and the validity of the resulting conclusions.

- **Unclear pseudo-label improvement:** The effect of continuous pseudo label generation on label quality over time is unreported; quantitative comparisons with ground truth are needed.

- **Limited modality evaluation:** SAM’s performance was only tested on colonoscopy images. Testing on more complex modalities, such as CT and MRI, would provide a fuller assessment of SAM’s versatility in medical imaging.

**Questions:**

1. A comparison between the pseudo labels generated by U-Net++ and those produced by SAM when using point and bounding box prompts would add depth to the evaluation. Since SAM was pre-trained exclusively on natural images, it may not inherently provide better pseudo labels than U-Net++, which has been directly trained on colonoscopy images. Quantifying the accuracy of each model’s pseudo labels against ground truth annotations would provide insight into their relative effectiveness.

2. When using point and bounding box prompts, did you introduce negative points or scribes to explicitly mark background areas? Including these background indicators would help SAM to distinguish between foreground and background more accurately, potentially improving the quality of the generated pseudo labels.

3. As Intersection over Union (IoU) and Dice metrics are mathematically related, reporting only one would be sufficient. To further enhance the evaluation, consider adding boundary-focused metrics, such as the Hausdorff Distance (HD), which provides a more precise measure of segmentation accuracy along object boundaries. This would give a more comprehensive assessment of the model’s performance in accurately delineating structures.

4. Continuous pseudo label generation lacks clarity. It is not reported whether the quality of pseudo labels improves over time compared to the ground truth. Given that randomly sampled points and bounding boxes are unlikely to change significantly, even as U-Net++ predictions improve slightly, substantial quality improvement seems unlikely. Ultimately, U-Net++’s pseudo labels might even surpass those generated by SAM. More quantified results are needed to monitor the label quality of both SAM and UNet++.

5. The SAM model was pre-trained on a large dataset of natural image masks, making it well-suited for tasks involving images similar to natural scenes. Given that colonoscopy images (in RGB format) resemble natural images in texture and color, SAM is likely beneficial for this application. However, to fully explore SAM’s versatility in medical imaging, it would be valuable to test its performance on more complex and unique modalities, such as CT and MRI scans, which differ significantly from natural images in terms of structure and contrast.

---

### Official Review · Reviewer_V2WM · 2024-11-04

**Soundness:** 3
**Presentation:** 2
**Contribution:** 2
**Rating:** 5
**Confidence:** 5

**Summary:**

The paper introduces a semi-supervised learning method that leverages the SAM to improve the performance of a smaller, task-specific model for medical image segmentation tasks. Specifically, the authors utilize SAM to generate pseudo labels for unlabeled data for further training of the smaller, task-specific model. The method demonstrates promising performance improvements on the two datasets.

**Strengths:**

1. The proposed method is reasonable and sound. Utilizing the prediction of the task-specific model as the prompt for SAM, better pseudo label can be obtained.
2. The authors validate their approach across two datasets, showing performance improvement. And the ablations show the effectiveness of each component and variants of the proposed method.

**Weaknesses:**

1. The major technical contribution is incremental, i.e., utilizing the task-specific model’s prediction as a prompt for SAM is straightforward.
2. Time and computational complexity. Although continuous pseudo-label generation outperforms the one-time approach, it introduces extra computational overhead. This could make the method less practical for large-scale deployment or time-sensitive applications.
4. Lack of comparison with respect to other semi-supervised segmentation methods.

**Questions:**

1. The proposed method leverages SAM for generating and refining pseudo-labels, though several of the semi-supervised techniques utilized are well established in conventional biomedical image segmentation tasks.

2. What are the time and computational costs associated with continuous pseudo-label generation? Additionally, what stopping criterion is applied in this setting?

3. The experimental setup lacks clarity. Could the authors explain the rationale for comparing the proposed method with other pre-training approaches? Also, how are the models trained under specific settings such as “SimCLR + continuous pseudo-labeling” and “MedSAM + continuous regularization”?

4. In Section 2.5, how was the lambda value determined?

5. Crucial baselines are missing: for example, training the model on 75% (or 50%) labeled data and using the small, task-specific model to generate pseudo-labels for training. The paper also lacks comparisons with other state-of-the-art semi-supervised methods.

---

### Official Review · Reviewer_ktkQ · 2024-11-04

**Soundness:** 2
**Presentation:** 2
**Contribution:** 2
**Rating:** 3
**Confidence:** 4

**Summary:**

This paper proposes a semi-supervised approach to medical image segmentation that uses knowledge from a large-scale model, SAM, to enhance a smaller U-Net++ model. By prompting SAM with outputs from U-Net++ on unlabeled images, SAM generates refined pseudo-labels, which are used to iteratively improve U-Net++ performance. The method is validated on Kvasir-SEG and COVID-QU-Ex datasets, where it demonstrates gains over baseline U-Net++ models trained only on labeled data.

**Strengths:**

1. The integration of pseudo-labels from SAM to iteratively enhance U-Net++ is implemented, merging pseudo-labeling with interactive training in a systematic manner that yields consistent improvements.
2. Although lacking innovation, the method demonstrates practical enhancements, rendering it potentially beneficial for medical environments with insufficient labeled data. The performance improvements demonstrate the method's effectiveness in scenarios with limited labels.
3. The paper  evaluates the method across prompt types and pseudo-label generation strategies, providing helpful insights into factors influencing segmentation quality.

**Weaknesses:**

1. Similar large-to-small model knowledge transfer techniques, including pseudo-labeling and iterative refinement, are already widely used in both medical and general image segmentation. Other work has applied large generalist models to medical tasks via pseudo-labeling pipelines (e.g., using complementary large models to generate pseudo-labels for smaller, specialized models). Although SAM is a newer model, the use of a large generalist model to guide smaller models lacks originality in the broader context of semi-supervised segmentation.
2.The accuracy of pseudo-labels generated by SAM is a critical factor. Poor initial predictions from U-Net++ could result in suboptimal prompts for SAM, which could subsequently generate flawed pseudo-labels, thereby degrading the learning process, if no additional filtering or uncertainty assessment is implemented.

**Questions:**

1. What makes this method different from other large-to-small model transfer techniques in medical image segmentation when it comes to knowledge mining or pseudo labeling? What makes it different from other approaches?
2.Could the authors talk about what SAM can't do when it comes to medical images, especially when it comes to structures with lots of small details or low contrast?

---

### Note · Authors · 2024-12-06

I have read and agree with the venue's withdrawal policy on behalf of myself and my co-authors.